# Marine Polyether Phycotoxin Palytoxin Induces Apoptotic Cell Death via Mcl-1 and Bcl-2 Downregulation

**DOI:** 10.3390/md21040233

**Published:** 2023-04-06

**Authors:** Jaemyun Kim, Seungwon Ji, Jin-Young Lee, Jean Lorquin, Barbora Orlikova-Boyer, Claudia Cerella, Aloran Mazumder, Florian Muller, Mario Dicato, Olivier Detournay, Marc Diederich

**Affiliations:** 1Department of Pharmacy, College of Pharmacy, Seoul National University, 1 Gwanak-ro, Gwanak-gu, Seoul 08626, Republic of Korea; vankim36@naver.com (J.K.); jsw0525@gmail.com (S.J.); barbora.orlikova@lbmcc.lu (B.O.-B.); claudia.cerella@lbmcc.lu (C.C.); aloran.mazumder@gmail.com (A.M.); 2Institut Méditerranéen d’Océanologie, 163 Avenue de Luminy, CEDEX 09, 13288 Marseille, France; 3Laboratoire de Biologie Moléculaire et Cellulaire du Cancer (LBMCC), BAM3 Pavillon 2, 6A Rue Nicolas-Ernest Barblé, L-1210 Luxembourg, Luxembourg; 4Planktovie SAS, 45 Rue Frédéric Joliot Curie, CEDEX 13, 13013 Marseille, France

**Keywords:** palytoxin, cell death, apoptosis, Bcl-2, Mcl-1, leukemia, solid tumors

## Abstract

Palytoxin is considered one of the most potent biotoxins. As palytoxin-induced cancer cell death mechanisms remain to be elucidated, we investigated this effect on various leukemia and solid tumor cell lines at low picomolar concentrations. As palytoxin did not affect the viability of peripheral blood mononuclear cells (PBMC) from healthy donors and did not create systemic toxicity in zebrafish, we confirmed excellent differential toxicity. Cell death was characterized by a multi-parametric approach involving the detection of nuclear condensation and caspase activation assays. zVAD-sensitive apoptotic cell death was concomitant with a dose-dependent downregulation of antiapoptotic Bcl-2 family proteins Mcl-1 and Bcl-xL. Proteasome inhibitor MG-132 prevented the proteolysis of Mcl-1, whereas the three major proteasomal enzymatic activities were upregulated by palytoxin. Palytoxin-induced dephosphorylation of Bcl-2 further exacerbated the proapoptotic effect of Mcl-1 and Bcl-xL degradation in a range of leukemia cell lines. As okadaic acid rescued cell death triggered by palytoxin, protein phosphatase (PP)2A was involved in Bcl-2 dephosphorylation and induction of apoptosis by palytoxin. At a translational level, palytoxin abrogated the colony formation capacity of leukemia cell types. Moreover, palytoxin abrogated tumor formation in a zebrafish xenograft assay at concentrations between 10 and 30 pM. Altogether, we provide evidence of the role of palytoxin as a very potent and promising anti-leukemic agent, acting at low picomolar concentrations in cellulo and in vivo.

## 1. Introduction

Palytoxin (Figure 1) was initially found in soft corals from tropical areas of the Pacific Ocean, for example, Hawaii; it was first discovered in the tropical soft coral Palythoa species, a zoanthid [1]. Palytoxin was first characterized by the chemist Moore in 1981. This compound represents one of the largest polyether-type phycotoxins [1]. This non-peptidic toxin consists of a long polyhydroxylated and partially unsaturated chain of 129 carbon atoms, and the aliphatic backbone contains 64 chiral centers [2]. Eight double bonds can exhibit geometrical *cis/trans* isomerism so that palytoxin can present over 1021 stereoisomers [3]. Structural analogs of palytoxin were also discovered and included ostreocin [4], ovatoxin [5], and mascarenotoxin [6]. The molecular weight of palytoxin ranges from 2658 to 2680 g/mol, depending on the Palythoa species from which it was obtained [2]. Palytoxin features lipophilic and hydrophilic regions and is considered a super-carbon-chain compound with the longest chain of carbon atoms in any known natural product [7]. Palytoxin is heat-stable, resists boiling, and remains stable in neutral aqueous solutions. A rapid decomposition is observed under acid or alkaline conditions leading to the loss of its toxicity [2].

The molecular action of palytoxin leads to the inhibition of the Na^+^/K^+^ adenosine triphosphatase (ATPase) channel [8,9]. This transmembrane protein exchanges three Na^+^ ions and two K^+^ ions using ATP hydrolysis as the driving force. The electrochemical gradient generated by the sodium pump for maintaining cell homeostasis is then inhibited by palytoxin. Palytoxin binds to the extracellular domain of the Na^+^/K^+^-ATPase to transform the pump into a non-specific, permanently open ion channel causing membrane depolarization, voltage-gated ion channel opening, and a massive increase of calcium ions (Ca^2+^) into the cytosol [10] strongly interfering with vital functions of the cell. An alteration of intracellular cation concentration, in particular, a calcium increase, is associated with cell death as a modification in the distribution of these ions across the plasma membrane plays a crucial role in palytoxin-induced cell-type specific cytotoxic response [11]. Further characterization of the anticancer mode of action of this compound would allow a better understanding of cell death mechanisms triggered by palytoxin at the cellular level. 

Here, we first focused on assessing differential cell toxicity and mechanisms involved in palytoxin-induced cell death in human leukemia cell lines. Palytoxin activates intrinsic and extrinsic apoptotic pathways triggered by alterations in the expression levels of the pro- and antiapoptotic Bcl-2 family proteins. Moreover, we provide evidence that palytoxin inhibits 3D tumor growth in colony formation assays and xenograft models. Altogether our results show that palytoxin acts as a potent and promising anti-leukemic agent at picomolar concentrations.

## 2. Results

### 2.1. Cytotoxic Effect of Palytoxin on Human Cancer Cells

Before investigating the cell death mechanisms, we first evaluated the antiproliferative and cytotoxic effects of palytoxin on various leukemia (Figure 2A–C) (Appendix A) and solid tumor cell lines (Appendix A). Cells were treated for 2, 4, 6, and 8 h at different concentrations of palytoxin, and viability was assessed by a Trypan Blue exclusion assay. Our results showed that solid tumor cell lines were more resistant to palytoxin under our experimental conditions than leukemia cell types. We also used PBMCs from healthy donors (Figure 2D) to evaluate the differential toxicity of palytoxin. We treated cells with palytoxin at various concentrations and conducted Trypan Blue staining to assess cell integrity and viability. At concentrations of 100 pM, palytoxin did not generate levels of toxicity comparable to cancer cells.

### 2.2. Effect of Palytoxin on Healthy Zebrafish Embryos

Palytoxin was used at various concentrations to treat zebrafish embryos and larvae for 24 h. Interestingly, we did not observe any toxicity nor alterations of the morphology of the yolk-sac nor axial/tail malformations (Figure 3A–C). Altogether this compound demonstrates a remarkable differential toxicity and selectively targets cancer cells.

### 2.3. Palytoxin-Induced Cell Death Leads to Caspase Activation in U937 Cells

We subsequently investigated whether palytoxin could trigger caspase-dependent apoptotic cell death. Palytoxin-induced cell death was identified by changes in the nuclear morphology typical of apoptotic cell death after staining with Hoechst and PI (Figure 4A). As caspase activation is sensitive to pan-caspase inhibitor zVAD, we concluded that palytoxin induces apoptosis (Figure 4B). Palytoxin induced the cleavage of the pro-caspases-8 and -9 and executor pro-caspases-3 and -7 after 6 h of treatment in a dose-dependent manner (Figure 4C). These results were further confirmed by luminescent assays where palytoxin induced the enzymatic activity of caspase-3/7 in U937 cells (Figure 4D).

### 2.4. Palytoxin Downregulates Expression of Antiapoptotic Bcl-2 Family Proteins

As the induction of apoptosis can be triggered by the inhibition of antiapoptotic proteins or activation of proapoptotic mechanisms, we investigated the overall expression levels of the Bcl-2 family proteins after palytoxin treatment (Figure 5A). Myeloid cell leukemia-1 (Mcl-1) and B-cell lymphoma-extra large (Bcl-xL) expression are downregulated in a dose-dependent manner. In contrast, B-cell lymphoma 2 (Bcl-2) is affected to a lesser extent. Moreover, proapoptotic Bcl-2 interacting domain death agonist (BID) was truncated by palytoxin treatment (Figure 5B), which implied that the amplification of intrinsic and extrinsic cell death pathways could be expected [12]. We generalized our finding by investigating the effect of palytoxin on a range of leukemia cell types (Figure 5C–E). Our results show a consistent decrease in Mcl-1 expression before the execution of cell death and before any other Bcl-2 family protein. We further generalized our findings by validating the modulation of Bcl-2 family proteins in solid tumor cell lines (Appendix A).

### 2.5. Mcl-1 Is Downregulated by Palytoxin in a Proteasome-Dependent Manner 

It is well established that antiapoptotic Mcl-1 can be downregulated by several mechanisms [13], including caspase-mediated- or proteasome-dependent degradation [14]. Accordingly, we assessed palytoxin-induced Mcl-1 degradation in the presence of two prototypical inhibitors, zVAD and MG-132. Our results show that palytoxin-induced Mcl-1 degradation is completely abolished in the presence of MG-132, underlining the essential role of proteasome degradation in the effect of palytoxin (Figure 6A). Unexpectedly, the three major proteasomal enzymatic activities, including trypsin, chymotrypsin, and caspase-like, were upregulated after palytoxin treatment, most likely further exacerbating the degradation of Mcl-1 (Figure 6B).

To get a deeper insight into the chronology of the molecular events triggered by palytoxin, we attempted to identify drivers of cell death rather than investigate the consequence of ongoing cellular demise. We used a kinetic approach to compare proteasomal activation, Mcl-1 degradation, and caspase activation. We obtained a rapid and significant activation of proteasomal activities before Mcl-1 degradation and caspase cleavage (Figure 6C). In parallel, ATP levels were quantified in the cells. We also observed a rapid decline in ATP levels (Figure 6D). BID was cleaved at early steps, which is in line with our hypothesis, as BID requires polyubiquitination and proteasomal activation for its processing (Figure 6D). Altogether, we concluded that palytoxin selectively targets essential cell death regulators before activation of the executioner caspase 3 and overall cellular degradation.

### 2.6. Bcl-2 Serine 70 Dephosphorylation Induced by Palytoxin Is Mediated through Protein Phosphatase 2A Activation

Our results also show that palytoxin affects Bcl-2 expression and phosphorylation levels witnessed by a discrete upshift of the band [15] (Figure 7A). We investigated phosphorylation levels of Bcl-2 as it is well known that serine 70 phosphorylation is required for the pro-survival function of this protein [16]. Our results show a dose-dependent dephosphorylation of Bcl-2, potentially contributing to a loss of its pro-survival function (Figure 7A). Serine 70 phosphorylation of Bcl-2 can be abrogated by phosphatase activation, mainly by protein phosphatase 2A (PP2A) activity [17]. Accordingly, in the next step, we used chemical inhibitors of selected cell signaling pathways, including okadaic acid used at a concentration inhibiting specifically PP2A. Our results show that PP2A inhibition partially rescues U937 cells from palytoxin-induced cell death, whereas other kinase pathways do not seem to be involved. (Figure 7B).

Altogether, we concluded that palytoxin-induced dephosphorylation of Bcl-2 by PP2A exacerbates the proapoptotic effect of Mcl-1 and Bcl-xL degradation and that okadaic acid could rescue in part palytoxin-induced cell death.

### 2.7. Anticancer Effect of Palytoxin in 3D Cell Culture and Zebrafish Xenografts

We investigated the colony formation ability of cancer cells in the presence of increasing concentrations of palytoxin to confirm its potential to impair the replicative capacity of cancer cells. After 10 days of treatment, colony formation in AML cell lines was strongly reduced by palytoxin at concentrations from 0.4 pM. Concentrations beyond 1 pM completely abrogated colony formation capacity (Figure 8A–D). Besides AML cells, palytoxin significantly inhibited colony formation in CML cell lines, K562 and imatinib-resistant K562 (K562IR) (Figure 8E,F). Moreover, we generalized our findings with the solid prostate cancer cell line PC3. Concentrations between 0.5 and 1 pM abrogated colony formation (Appendix A) and prevented spheroid formation in a dose-dependent manner (Appendix A).

To further extend our evaluation of the anticancer potential of palytoxin, we examined the ability of palytoxin to abrogate tumor development in an ex vivo zebrafish xenotransplantation model. Fluorescently labeled leukemia cells were pre-treated for 6 hours with palytoxin at 5, 10, and 30 pM and then injected into the yolk sac of zebrafish embryos. Results revealed that the tumor-associated fluorescence intensity signal was drastically lowered in the palytoxin-treated zebrafish group compared to the untreated controls as early as 5 pM (Figure 9A–F for a representative overview; Appendix A shows all nine fish for each group).

## 3. Discussion

We demonstrated that palytoxin, one of the most toxic marine biotoxins known [18], can significantly induce apoptotic cell death at low picomolar concentrations in different cancer cell types. We focused on cell death mechanisms, especially in acute myeloid leukemia (AML) U937 cells. 

So far, palytoxins have essentially been investigated at high nanomolar doses. Palytoxins act as a skin tumor promoter when combined with a carcinogenic agent. It can modulate critical signal transduction pathways involved in carcinogenesis and inflammation [19]. It has been shown that palytoxin stimulates prostaglandin production from arachidonic acid and activates mitogen-activated protein kinases (MAPKs), including extracellular signal-regulated kinase (ERK), c-Jun N-terminal kinase (JNK) and p38 mitogen-activated protein kinase (p38) [20].

In contrast, this manuscript shows that palytoxin induced apoptosis at lower pM concentrations in a caspase-dependent manner. Apoptosis is the most common and well-defined form of programmed cell death, a physiological process of cellular suicide [21]. However, as we observed a small measurable amount of non-apoptotic cell death after caspase inhibition by zVAD, it is essential to determine in the future which type of secondary cell death mechanism the cell activates in the absence of apoptotic capacity [22]. In that sense, investigating the activation of non-caspase proteases inducing calpains or cathepsins could be interesting [23,24].

Among Bcl-2 family proteins, Mcl-1 and Bcl-xL were found to be strongly downregulated by palytoxin, even at low picomolar concentrations. As we observed a consistent and early onset of BID truncation, the differential effect of extrinsic versus intrinsic cell death pathway is interesting and will be investigated by pathway-specific caspase inhibitors [12].

We demonstrated that palytoxin rapidly reduces Mcl-1 protein levels without involving caspase-dependent modulation. Palytoxin instead induced proteasome-dependent degradation of Mcl-1. Three major proteasome activities are upregulated by palytoxin. Ubiquitin E3 ligases Skp Cullin F-box containing F-box WD repeat domain containing 7 (SCFFbw7), β-transducin repeat-containing protein 1 (SCFβ-TrCP), and Mcl-1 ubiquitin ligase E3 (Mule) [25,26] could then promote Usp9x (ubiquitin-specific peptidase 9, X-linked) which is required for Mcl-1 deubiquitination [27]. Moreover, Bcl-2 homology domain-3 (BH-3) only proteins, including p53 upregulated modulator of apoptosis (PUMA) or Phorbol-12-myristate-13-acetate-induced protein 1 (NOXA), induce the degradation of Mcl-1 at the mitochondrial level, which also requires the E3 ligase Mule [28]. The implication of NOXA in regulating Mcl-1 proteasomal degradation will support our hypothesis as the disruption of the 26S proteasome function by different mechanisms triggers the rapid accumulation of proteasome-dependent BH-3-only family proteins and subsequent cell death induced by apoptosis [28].

It will be interesting to investigate the overexpression of Mcl-1 with plasmid constructs, specifically wild-type and proteasome-resistant isoforms of Mcl-1 [29]. Overexpression of these proteins will further allow the strengthening of the hypothesis of proteasomal degradation. Similarly, stably-transfected Jurkat T cells expressing mutated Bcl-2 (serine70, serine87, and threonine 69) will allow us to understand the role of Bcl-2 expression regulated by palytoxin in various models [30,31].

This study also showed that inhibition of PP2A activity rescued palytoxin-induced cell death, which implies that phosphatases play a significant role in cellular mechanisms activated by palytoxin [32]. PP2A has been suggested as a therapeutic target in Philadelphia chromosome-positive chronic or acute myeloid leukemia, where PP2A inactivation is a recurrent event [32]. Palytoxin could restore PP2A or related phosphatases to induce caspase-dependent apoptosis mediated through AKT and ERK1/2 pathways [33]. PP2A exerts inhibitory and stimulatory effects on MAP kinase signaling pathways [34]. Previous findings suggested that palytoxin activates the mitogen-activated protein kinase (MAPK) cascade (including extracellular signal-regulated protein kinases 1 and 2 (ERK1/2), c-Jun N-terminal kinases/stress-activated protein kinase (JNK) and p38 protein kinase) in response to heat shock protein 27 (HSP27) phosphorylation in human breast adenocarcinoma (MCF-7) cells [35]. In monocyte-derived human macrophages, palytoxin phosphorylates p38 MAPK. It activates nuclear factor kappa B (NF-κB) pathways by increasing the mRNA levels of inflammation-related genes like interleukin 8 (IL-8), an inhibitor of kappa alpha (IκB-α), leading to NF-κB nuclear translocation [6]. These observations allow us to hypothesize that palytoxin may activate a proinflammatory signaling cascade in leukemia cell lines by regulating kinases and phosphatases [36]. 

This manuscript is the first step in the preclinical evaluation of palytoxin. We are aware that there might be a long way to derive this high molecular weight compound into a druggable formulation. However, the potential of Na^+^/K^+^ ATPase modulating agents was clinically investigated and exploited by cardiac glycosides so that the molecular target of palytoxin is of clear clinical relevance. In the future, the potent cytotoxicity, combined with the described proinflammatory potential, could find clinical applications in targeted antibody-drug conjugate (ADC) compounds like brentuximab vedotin or trastuzumab emtansine. Despite the limited success of first-generation ADCs, palytoxin could provide a cytotoxic, likely immunogenic warhead able to kill difficult-to-target cancer cells.

One of the future outcomes of this study is to validate the Na^+^/K^+^-ATPase inhibition by palytoxin as the possible cancer target. In this context, functional studies (i.e., silencing or overexpressing the pump subunits) and the future design of simplified analogs based on essential pharmacophores of palytoxin might be considered. Palytoxin inhibited tumor formation in a zebrafish xenograft model at concentrations between 10 and 30 pM. We plan to extend our investigation to animal models other than zebrafish to exclude any intrinsic evolutionary resistance developed by aquatic animal models. Furthermore, studies on primary cells will consolidate the documented differential cytotoxicity.

## 4. Materials and Methods 

### 4.1. Cells and Medium

K562 (human chronic myelogenous leukemia), Jurkat (human T-lymphocyte), Raji (Burkitt lymphoma), and U937 (human histiocytic lymphoma) were purchased from DSMZ (Braunschweig, Germany). THP1, promyeloblast HL60 and bone marrow acute myelogenous leukemia KG-1 were obtained from the Korean cell line bank (KCLB, Seoul, Republic of Korea). All cells were cultured in Roswell Park Memorial Institute (RPMI) 1640 medium (Bio-Whittaker, Lonza, Rockland, ME, USA) containing a 1% (*v*/*v*) antibiotic/antimycotic mixture of 100 U/mL penicillin, 100 μg/mL streptomycin, and 0.25 μg/mL amphotericin B (Bio-Whittaker, Lonza, Rockland, ME, USA) at 37 °C and 5% of CO_2_ in a humidified atmosphere. The Imatinib-resistant K562 (K562IR) cell line was a gift from the Korea Leukemia Bank, Catholic University, Seoul, and cultured in RPMI medium with 25 mM HEPES (Lonza, Rockland, ME, USA) supplemented with 10% (*v*/*v*) FBS, 1% (*v*/*v*) antibiotic-antimycotic and 1 μM of imatinib. Cells were cultured and harvested every three days and treated with palytoxin in an exponential growth phase. Cells were regularly tested for mycoplasma infection (MycoalertTM, Lonza, Rockland, ME, USA) per the manufacturer’s instructions.

### 4.2. Compounds

Palytoxin was extracted from Palythoa Aff. Clavata following an established procedure (patent publication EP3087172B1). The compound, with a molecular mass of 2680.14 g mol^−1^, was received as a powder, solubilized in DMSO (Sigma-Aldrich, Saint Louis, MO, USA), and further diluted to get working aliquots at 1 mM. Stocks and aliquots were stored at 4 °C for up to 2 years without loss of activity and protected from light and were used directly before the experiments. Etoposide was purchased from Sigma-Aldrich, Saint Louis, MO, USA, and dissolved in DMSO at a stock concentration of 50 mM. Pan-caspase inhibitor, z-VAD FMK, was purchased from Calbiochem (San Diego, CA, USA), dissolved in DMSO, and added 1 h before at a concentration of 50 μM. Protein phosphatase 2A inhibitor okadaic acid was purchased from Calbiochem (San Diego, CA, USA) and dissolved in DMSO at a stock concentration of 1 mM. Proteasome inhibitor MG-132 was purchased from Sigma-Aldrich (Saint Louis, MO, USA) and dissolved in DMSO at a stock concentration of 10 mM. MAP kinase inhibitors, SB202190 and PD98059, were purchased from Calbiochem (San Diego, CA, USA) and dissolved in DMSO at a stock concentration of 100 mM. Hydroquinone was purchased from Sigma-Aldrich (Saint Louis, MO, USA) and dissolved in DMSO at 10 mM. Inhibitors were used for 1h before palytoxin treatment at the indicated working concentrations.

### 4.3. Cell Viability Assessment

Cell vitality was assessed in three leukemia cell lines, K562, Jurkat, and U937. Each cell line was treated with various concentrations of palytoxin for 2, 4, 6, and 8 h in a dose-dependent manner. The percentage of viability was evaluated using the Trypan Blue exclusion test (0.2% Trypan Blue), and the percentage of viability of cells without palytoxin treatment was considered to be 100%.

### 4.4. Caspase 3/7 Activity Assay

3 × 10^5^ U937 cells/mL were seeded in 96-well plates in triplicate in 75 μL for palytoxin treatment. The enzymatic activities of caspases- 3 and -7 were determined using a luminescent Caspase-glo®3/7 Assay (Promega, Cosmogenetech, Seoul, Republic of Korea). The assay was performed per the manufacturer’s instruction by adding 75 μL of the caspase-3/7 substrate to U937 cells in suspension. After one hour of incubation with DEVD-aminoluciferin substrate and luciferase, the luminescent signal was measured using a Centro LB 960 microplate luminometer (Berthold Technologies, JCBio Co. Ltd., Seoul, Republic of Korea).

### 4.5. Proteasome Activity Assay

The Proteasome-Glo® cell-based assay (Promega, Cosmogenetech, Seoul, Korea) was used to evaluate the three major proteolytic enzyme activities (chymotrypsin-like, caspase-like, trypsin-like). The assays were performed as indicated in the manufacturer’s protocol. U937 cells were treated at a concentration of 10^6^ cells/mL in RPMI 1640 medium containing 0.1% FCS at indicated concentrations of palytoxin. After an incubation period, 50 μL of U937 cellular suspension was mixed with 50 μL of the assay reagent. The luminescence signal was measured on the Centro LB 960 Microplate Luminometer (Berthold Technologies, JCBio Co., Ltd., Seoul, Republic of Korea). The proteasome signal was normalized to the number of viable cells performed in parallel using CellTiter-Glo® (Promega, Cosmogenetech, Seoul, Republic of Korea). MG-132 at 5 μM (Sigma, USA) was used as a control for proteasomal inhibition.

### 4.6. Cell Lysate Preparations and Western Blots

Ten million U937 cells were seeded in 30 mL of RPMI 1640 medium (Lonza, Rockland, ME, USA) with 10% fetal bovine serum (FBS) (Biowest, Riverside, MO, USA) and 1% antibiotics (Lonza, Rockland, ME, USA). Cells were treated with palytoxin in a dose- and time-dependent manner as indicated. Cellular lysates were centrifuged at 22 °C, 350 g, for 7 min. After removal of the supernatant, the pellets were washed in 1ml phosphate-buffered saline (PBS) and centrifuged again at 4 °C, 350 g for 7 min. Afterward, the supernatants were removed, and the pellets were stored at −80 °C until use. The extraction was performed on ice to avoid the denaturation of the proteins. Whole-cell extracts were prepared with 5 mL of mammalian protein Extraction reagent (M-PER), per the manufacturer’s instructions (Pierce, Appleton, WI, USA). Extraction reagents include 40 μL/mL protease inhibitor (Sigma, USA), 1 mM phenylmethylsulphonyl fluoride (PMSF) (Roche, Seoul, Republic of Korea), 1 mM sodium orthovanadate (Saint Louis, MO, USA), 100 μL/mL Phosphostop inhibitor (Roche, Seoul, Republic of Korea) and M-PER in a final volume of 5 mL. Depending on the pellet size, 500 μL of reagents were used for control and 300 μL for treated cell pellets. Dissolved pellets were centrifuged for 25 min at 4 °C and 18,000 g speed. Aliquots were stored at −80 °C.

Proteins from total extracts were separated using a sodium dodecyl sulfate-polyacrylamide gel electrophoresis (SDS-Page). Proteins were transferred onto a polyvinylidene fluoride (PVDF) membrane (GE Healthcare, Pittsburgh, PA, USA) and blocked with 5% non-fat milk or 5% bovine serum albumin (BSA) in PBS-Tween overnight. Equal loading of samples was controlled by using β-actin. Blots were incubated with primary antibodies: anti-β-actin (1/10000, A5441, Sigma-Aldrich, Saint Louis, MO, USA), anti-Caspase-3 (1/1000, 56053, Santa Cruz, Koram Biotech Corporation, Seoul, Republic of Korea), anti-Bcl-2 (1/2000, Merck Calbiochem, OP60, Seoul, Republic of Korea), anti-Bcl-xL (1/1000, BD Biosciences, 610212, Seoul, Republic of Korea), anti-Caspase-7 (1/1000, 9494) anti-Caspase-8 (1/1000, 9746), anti-Caspase-9 (1/1000, 9502), anti-Mcl-1 (1/1000, 4572), anti-BID (1/1000, 2002), anti-pBcl-2 ser70 (1/2000, 2827) (all Cell Signaling Technologies, Koram Biotech Corporation, Seoul, Republic of Korea). All antibodies were diluted in a PBS-Tween solution containing 5% BSA or 5% non-fat milk per the provider’s protocols. After incubation with primary antibodies, membranes were washed for 3 × 10 min with PBS-Tween, followed by 1h at RT with the corresponding secondary (HRP-conjugated) antibodies. After washing 3 × 10 min with PBS-Tween, specific immunoreactive proteins were visualized by autoradiography using the ECL Plus Western Blotting Detection System Kit® (GE Healthcare, Seoul, Republic of Korea).

### 4.7. Fluorescent Microscopy Analysis

3 × 10^5^ U937 cells/mL were seeded in 24-well plates before palytoxin treatment. Then, 300 μL of treated cells were transferred into another 24-well plate for double staining with (1) the DNA-specific dye Hoechst 33342 (Sigma, USA) at 1 μL/mg for 30 min of incubation and (2) propidium iodide (Sigma, USA) at 1.5 μL/mg, diluted in PBS for 15 min of incubation. Induction of apoptosis was assessed by fluorescent microscopy (Nikon TI-U, Seoul, Republic of Korea) and expressed as the percentage of cells presenting fragmented/condensed nuclei. Different stages of nuclear fragmentation were considered, and at least 300 cells were counted in three independent fields as previously described [37,38,39,40,41,42]. Microscopy images were analyzed using the ImageJ 1.54 software (http://rsb.info.nih.gov/ij/docs/index.html, accessed on 4 March 2023).

### 4.8. Systemic Toxicity in Zebrafish

Zebrafish (Danio rerio) were obtained from the “Zebrafish International Resource Center” (ZIRC) (University of Oregon, Eugene, OR, USA) and maintained according to the zebrafish guidelines [43]. Adult fish were kept at 28.5 °C on a 14-h light/10-h dark cycle comparable to natural conditions, and all embryos were collected from natural mating as a unit of hours-post fertilization (HPF). Then, 0.003% phenylthiourea (PTU) was added 14 h before palytoxin treatment to remove the pigmentation of zebrafish models. Embryos were then treated with the indicated concentration of palytoxin in 24 HPF. Photographs were taken under light microscopy (Carl Zeiss Stereo microscope DV4, Seoul, Republic of Korea) to quantify viable zebrafish.

### 4.9. Differential Toxicity Effects on Healthy Peripheral Blood Mononuclear Cells

Peripheral blood mononuclear cells (PBMC) were purified from three freshly collected buffy coats from healthy adult human donors (Red Cross, Luxembourg, Luxembourg) after ethical approval (see informed consent statement) and informed written consent from all the donors, using the standard Ficoll-Hypaque (GE Healthcare, Roosendaal, The Netherlands) density separation method as previously described [40]. After isolation and three washes in Dulbecco’s Phosphate Buffered Saline (DPBS) (Lonza, Rockland, ME, USA), cells were counted and re-suspended in RPMI 1640 medium supplemented with 10% heat-inactivated fetal calf serum and 1% antibiotics-antimycotic at a cell density of 2 × 10^6^ cells/mL. The day after, cell concentration was adjusted to 1 × 10^6^ cells/mL, and PBMCs were treated with palytoxin at indicated concentrations. After 8 hours of incubation, a Trypan Blue assay was used to analyze PBMC cell viability.

### 4.10. Colony Formation Assay

For colony formation assays, cells (10^3^ cells/mL) were seeded into a semisolid methylcellulose medium (Methocult H4230, StemCell Technologies Inc., Seoul, Republic of Korea) and treated with indicated concentrations of palytoxin. Colonies were detected after 10 days of culture by adding 1 mg/mL of 3-(4,5-dimethylthiazol-2-yl)-2,5-diphenyltetrazoliumbromide (MTT) reagent (Sigma, Seoul, Republic of Korea) and were scored by ImageJ software (U.S. National Institute of Health, Bethesda, MD, USA) [44].

### 4.11. Zebrafish Xenograft Assays

The zebrafish (*Danio rerio*) xenograft assay followed a setup we previously published [45].

### 4.12. Statistical Analysis

Data are presented as the mean of three independent experiments with standard deviations. All statistical analyses were done with GraphPad Prism 9.0. *p*-values below 0.05 were considered significant. See figure legends for details.

## 5. Conclusions

We have demonstrated that the marine compound palytoxin rapidly induces cell death through apoptosis in human leukemia cell lines. While downregulating the expression of Mcl-1, palytoxin activates BID truncation, eventually leading to intrinsic apoptotic pathways are a caspase-dependent cell death mechanism. While additional analysis is required to elucidate the role of phosphatases that could be involved, these findings show for the first time that palytoxin acts as an apoptosis inducer affecting leukemia cells able to prevent tumor formation in vivo and makes palytoxin a promising candidate for new potential anticancer drug development.

## Figures and Tables

**Figure 1 marinedrugs-21-00233-f001:**
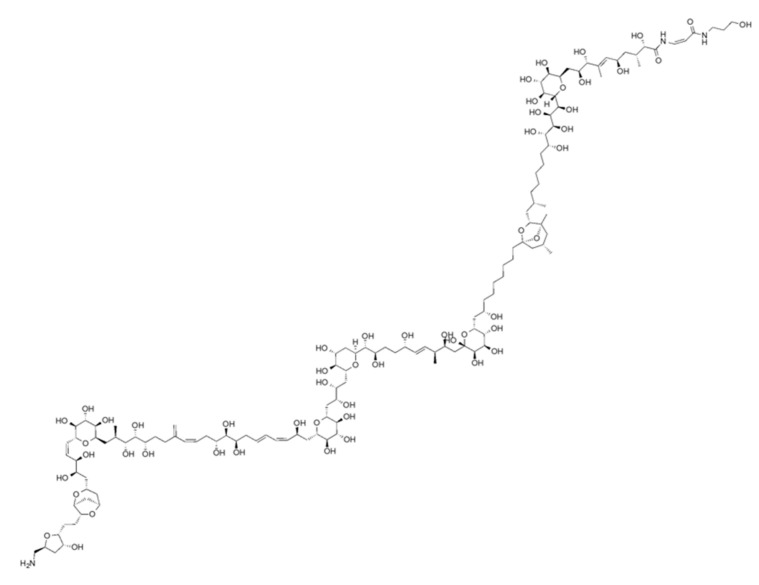
Structure of palytoxin.

**Figure 2 marinedrugs-21-00233-f002:**
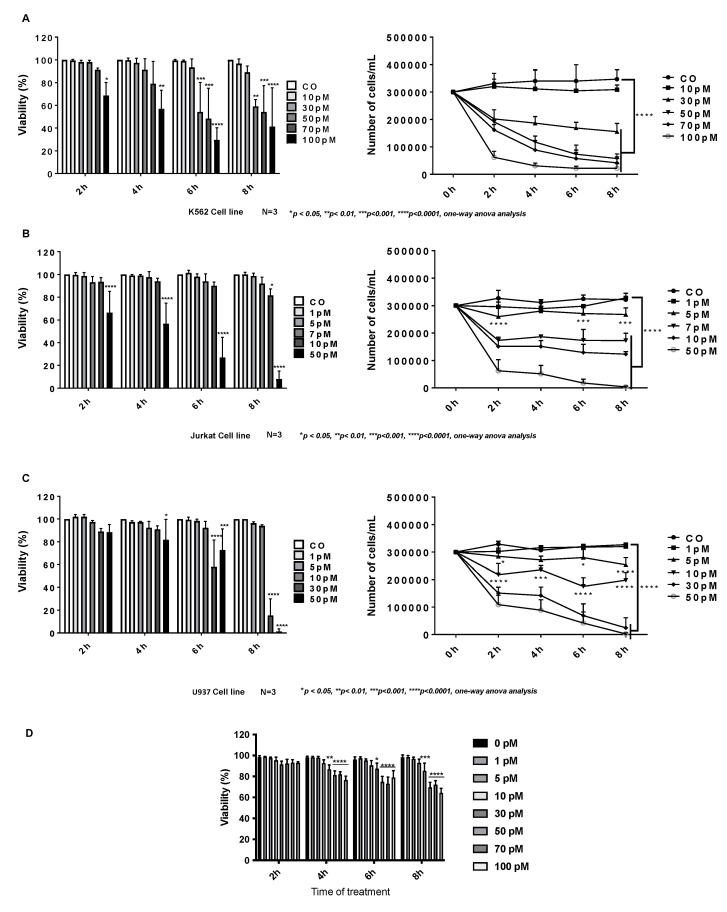
Differential cytotoxic effect of palytoxin on human leukemia cell lines. (**A**) Effects of Palytoxin on human chronic leukemia K562 cell viability and proliferation. (**B**) Effects of palytoxin on human T-cell leukemia Jurkat cell viability and proliferation. (**C**) Effects of Palytoxin on human histiocytic lymphoma U937 cell lines. (**D**) Time and concentration-dependent effect of palytoxin on the viability of healthy PBMCs from three buffy coats. Data are the mean of SD ± of three independent experiments. Statistical significance was assessed as * *p* < 0.05, ** *p* < 0.01, *** *p* < 0.001, **** *p* < 0.0001.

**Figure 3 marinedrugs-21-00233-f003:**
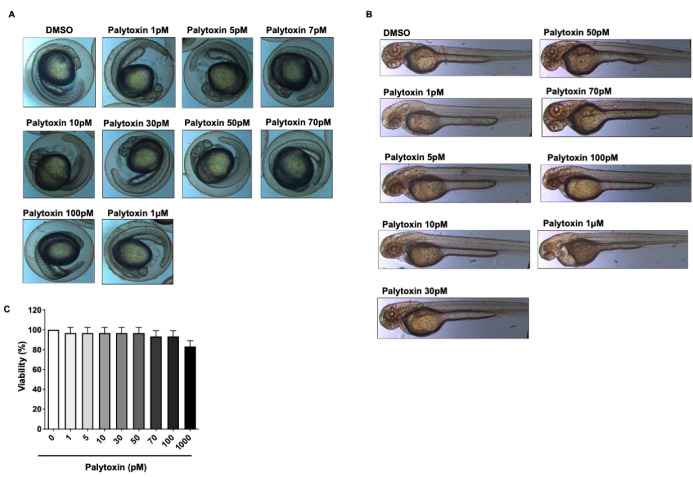
Differential effects of palytoxin on zebrafish embryos and healthy cells. (**A**) Effects of palytoxin on developmental defects in zebrafish embryos. Zebrafish embryos were incubated with palytoxin at concentrations from 1 pM to 1 μM for 24 h. (**B**) Effect of palytoxin exposure on zebrafish morphology scale bar: 20 μM. Zebrafish exposed to palytoxin did not exhibit moderate or severe defects of morphology. (**C**) Effect of palytoxin on the viability of healthy PBMCs. PBMCs from healthy donors were treated with indicated palytoxin concentrations for 24 h. Results correspond to the mean of ±SD of the quantification of three independent experiments.

**Figure 4 marinedrugs-21-00233-f004:**
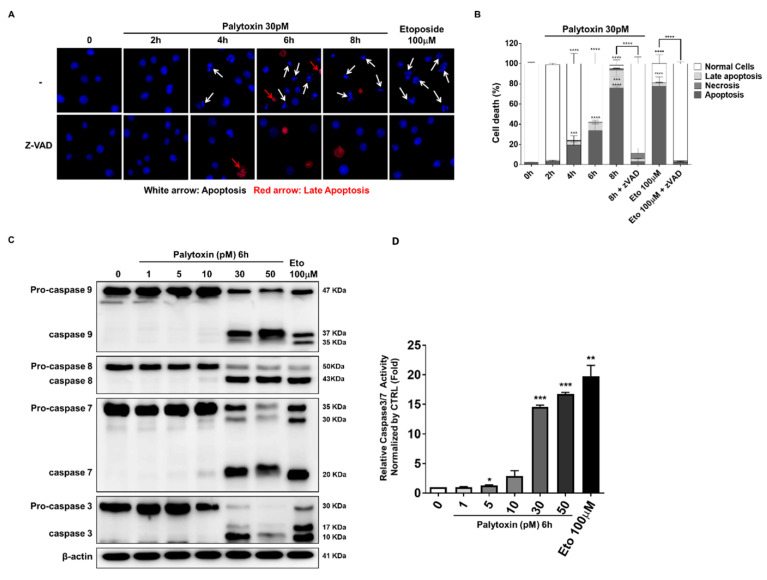
Palytoxin-induced cell death leads to caspase activation in U937 cells. (**A**) Hoechst/PI staining of palytoxin-treated cells. White arrows indicate cells displaying apoptotic features such as fragmented and condensed nuclei. Red arrows indicate PI-positive cells undergoing late apoptosis and presenting necrotic morphology. (**B**) The percentage of apoptotic cells was evaluated by counting cells corresponding to apoptosis, late apoptosis, and necrosis. At least 100 cells in three random fields were counted by fluorescence microscopy. (**C**) Palytoxin activation of pro-caspase-8, -9, -7, and -3 was investigated by Western blot analysis (left; one of three independent experiments). Enzymatic activity of caspases-3 and -7 was upregulated in a dose-dependent manner. (right; data representative of three independent experiments (mean ± SD). (**D**) Etoposide-treated cells (100 μM, 4 h) were positive controls for the induction of apoptosis. * *p* < 0.05, ** *p* < 0.01, *** *p* <0.001, **** *p* < 0.0001 versus control.

**Figure 5 marinedrugs-21-00233-f005:**
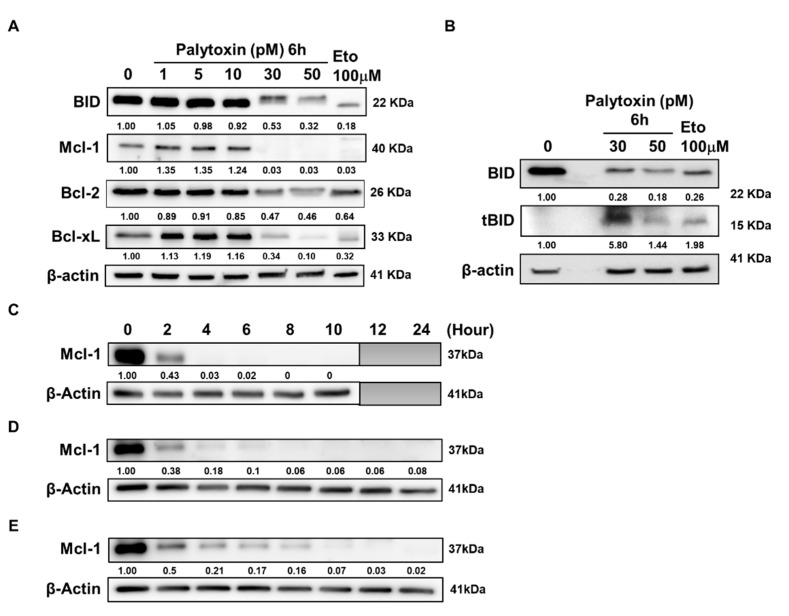
Palytoxin downregulates the expression of antiapoptotic Bcl-2 family proteins. (**A**) Western blot analysis of antiapoptotic Bcl-2 family protein expression levels inhibited by palytoxin after 6 h of treatment in U937. (**B**) Western blot analysis of the truncation of proapoptotic protein BID induced by palytoxin in U937. Western blot analysis of the Mcl-1 protein expression inhibited by palytoxin in a kinetic approach in K562 (**C**), Jurkat (**D**), and Raji (**E**). Western blot results are representative of three independent experiments.

**Figure 6 marinedrugs-21-00233-f006:**
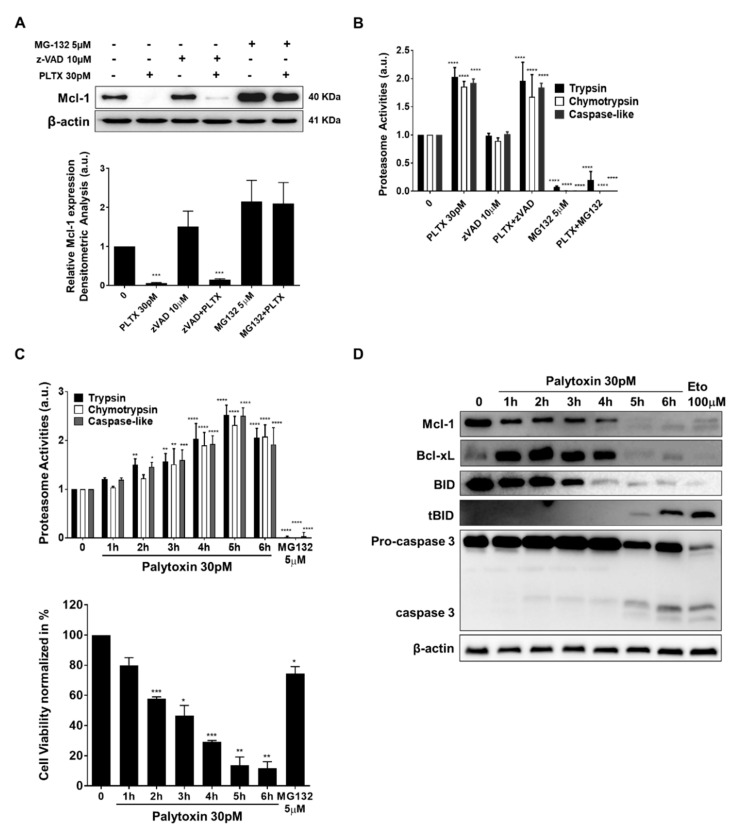
Mcl-1 is ubiquitously downregulated by palytoxin in a proteasome-dependent manner. (**A**) Palytoxin-induced proteasome-dependent degradation of Mcl-1 in U937 cells. Band intensities were quantified using ImageJ software. Values represent the mean of ±SD for three independent experiments. (**B**) The three major proteolytic activities of the 26S proteasome were assessed under the same conditions as the Western blots. MG-132 at 5 μM was used as a positive control for proteasomal inhibition in U937 cells. The asterisk indicates significant differences between treated versus control, **** *p* < 0.0001. (**C**) Kinetic analysis of the assessment of three major proteasome activities in U937 cells. Cell viability was determined to normalize the three major proteasome activities induced by palytoxin. (**D**) Expression of Mcl-1 and Bcl-xL degradation, BID truncation, and Caspase 3 cleavage was evaluated by Western blot analysis under the same conditions. Data representative of three independent experiments is shown as mean ± SD. * *p* < 0.05, ** *p* < 0.01, *** *p* < 0.001, **** *p* < 0.0001 versus control.

**Figure 7 marinedrugs-21-00233-f007:**
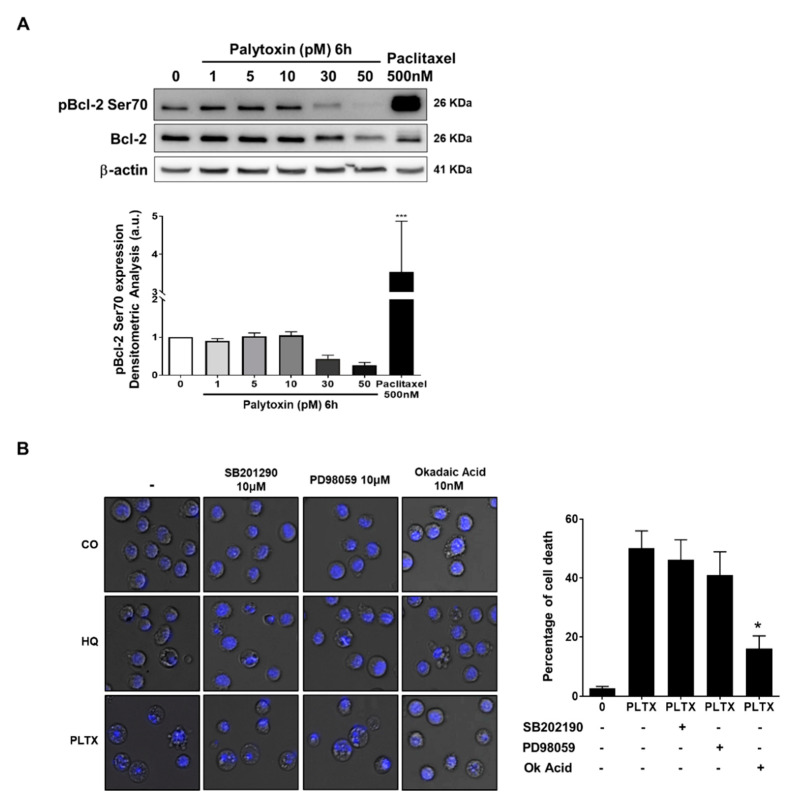
Bcl-2 serine70 dephosphorylation induced by palytoxin is mediated through protein phosphatase 2 activation in U937 cells. (**A**) Palytoxin induces a dose-dependent dephosphorylation of Bcl-2 at serine70 in U937 cells. Western blot is representative of three independent experiments. Band intensities were quantified using ImageJ software. Paclitaxel (500 nM, 8 h) is used as a positive control for multisite phosphorylation of Bcl-2. *** *p* < 0.005 versus control. (**B**) Okadaic acid, a protein phosphatase 2A (PP2A) inhibitor, rescues U937 cells from palytoxin-induced cell death. Hydroquinone (10 μM, 24 h) was used as a positive control for PP2A activation. Cells were stained with Hoechst to analyze the nuclear morphology merged with phase contrast images using ImageJ. The percentage of cell death was evaluated by counting the number of cells undergoing various types of cellular death. Three independent fields of 100 cells were quantified. SB202190 and PD98059 were used as MAP kinase inhibitors. Data representative of three independent experiments is shown as mean ± SD. * *p* < 0.05.

**Figure 8 marinedrugs-21-00233-f008:**
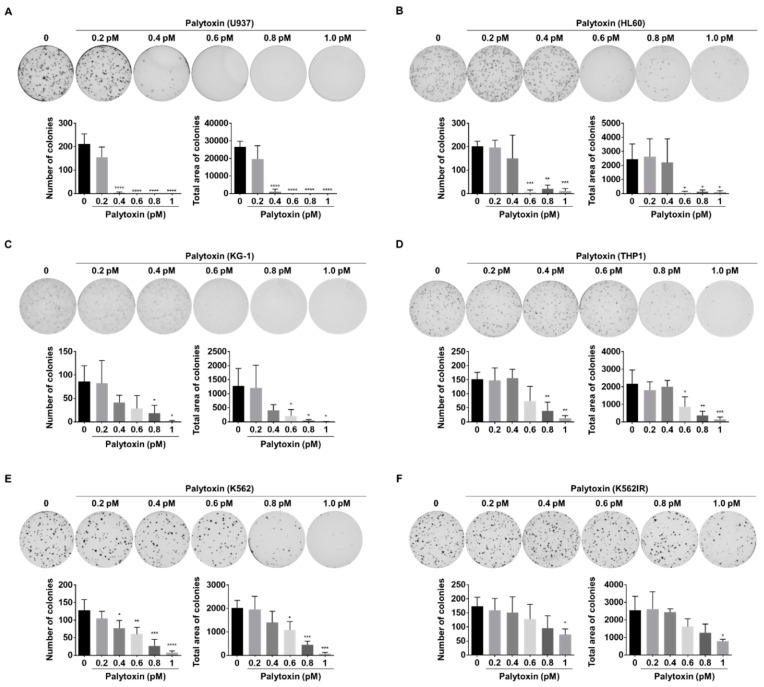
Palytoxin reduces the colony-forming ability in leukemia cell lines. Palytoxin showed an inhibitory effect on colony formation in a dose-dependent manner in U937 (**A**), HL60 (**B**), KG-1 (**C**), THP1 (**D**), K562 (**E**), and K562IR (**F**). All data represent the mean ± SD. of three independent experiments. Statistical significance was assessed as * *p* < 0.05, ** *p* < 0.01, *** *p* < 0.001, **** *p* < 0.0001 versus control.

**Figure 9 marinedrugs-21-00233-f009:**
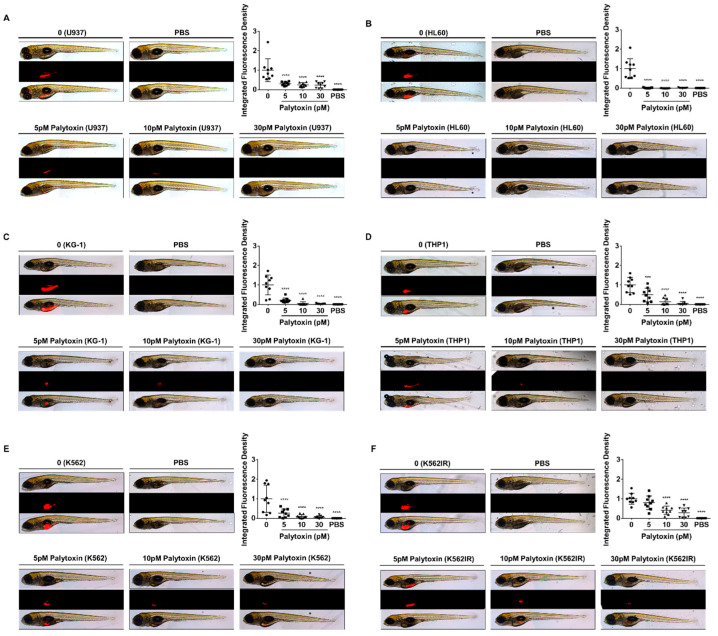
Palytoxin inhibits colony-forming ability in in vivo zebrafish xenografts. In vivo antitumor forming ability by palytoxin was evaluated by zebrafish xenografts. Top: bright field, middle: CM-Dil, bottom: merge. (**A**) U937, (**B**) HL60, (**C**) KG-1, (**D**) THP1, (**E**) K562, (**F**) K562IR cells were used for generating zebrafish xenograft and nine zebrafishes were assessed per condition (Appendix A). Statistical significance was assessed as *** *p* < 0.001, **** *p* < 0.0001.

## Data Availability

The data presented in this study are available on request from the corresponding author.

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
