# Peer review of "Marine Polyether Phycotoxin Palytoxin Induces Apoptotic Cell Death via Mcl-1 and Bcl-2 Downregulation"

_marinedrugs, 2023, doi:10.3390/md21040233_

Round 1

Reviewer 1 Report

In this manuscript, Kim et al. showed cytotoxic activities of palytoxin in several cancer cells and proposed its mechanism of action. Authors claimed that palytoxin down-regulated anti-apoptotic Bcl-2 family proteins Mcl-1 and Bcl-xL, thereby inducing apoptosis in leukemia cells. In addition, authors found the involvement of PP2A phosphatase in palytoxin-induced apoptosis. Finally, authors showed anti-proliferative activities of palytoxin in a 3D-culture model and a zebrafish xenograft model. The experiments appear to be well executed with suitable controls and results are enough to support conclusion. I would like to recommended it for publication in Marine drugs if several minor changes are applied.

Minor comments.

1.     In Fig 2D. instead of % cell death, cell viability should be measured in heathy PBMC.

2.     Authors need to discuss the possible toxicity (side effects) of palytoxin.

3.     In addition, authors need to highlight the possible clinical significance of their findings.

Author Response

Reviewer 1

In this manuscript, Kim et al. showed cytotoxic activities of palytoxin in several cancer cells and proposed its mechanism of action. Authors claimed that palytoxin down-regulated anti-apoptotic Bcl-2 family proteins Mcl-1 and Bcl-xL, thereby inducing apoptosis in leukemia cells. In addition, authors found the involvement of PP2A phosphatase in palytoxin-induced apoptosis. Finally, authors showed anti-proliferative activities of palytoxin in a 3D-culture model and a zebrafish xenograft model. The experiments appear to be well executed with suitable controls and results are enough to support conclusion. I would like to recommended it for publication in Marine drugs if several minor changes are applied.

We thank the reviewer for the overall positive evaluation of our manuscript.

  1. In Fig 2D. instead of % cell death, cell viability should be measured in heathy PBMC.

We thank the reviewer for this remark. Now, Figure 2D represents the results as “Cell viability (%)”.

  1. Authors need to discuss the possible toxicity (side effects) of palytoxin.

We thank the reviewer for this important comment. At high nanomolar doses, palytoxins can trigger inflammation, act as a tumor co-promotor, and contribute to carcinogenesis via MAP kinase signaling. We mention these effects in the discussion to contrast the compound's effect at high vs. low doses. Indeed, the aim of this manuscript was to provide a proof-of-principle for the anticancer potential of palytoxin at low picomolar concentrations. At these concentrations, the compound triggers apoptotic cell death mechanisms in both hematological and solid tumor cell models.

We adapted the introduction and discussion sections to match the text with our intentions and focus on the results of the present manuscript.

  1. In addition, authors need to highlight the possible clinical significance of their findings.

We agree with the reviewer and added the following information to the discussion section:

“This manuscript is the first step in the preclinical evaluation of palytoxin. We are aware that there might be a long way to derive this high molecular weight compound into a druggable formulation. Yet the potential of Na+/K+ ATPase modulating agents was clinically investigated and exploited by cardiac glycosides so that the molecular target of palytoxin is of clear clinical relevance. In the future, the potent cytotoxicity, combined with the described proinflammatory potential, could find clinical applications in targeted antibody-drug conjugate (ADC) compounds like brentuximab vedotin or trastuzumab emtansine. Despite the limited success of first-generation ADCs, palytoxin could provide a cytotoxic, likely immunogenic warhead able to kill difficult-to-target cancer cells”.

Reviewer 2 Report

Authors did a good job in making a good and important story about the the polyether phycotoxin palytoxin. The study is relevant to marine natural products and aims to highlights the differential and mechanistic cytotoxicity of title molecule against several AML and solid tumors.

Authors, please consider changing the title by replacing the word "compound" with "polyether phycotoxin", explicitly describing the chemical class of the title marine natural product. Authors, please change "phosphorylation" in line 35 to "dephosphorylation". I suggest describing the zebrafish colonization model as ex-vivo rather than in vivo since the treatment conducted outside the animal and not direct systemic treatment after xenografting. Line 49, please make cis/trans in italics font and add the word "geometrical" before them. I suggest to make the word "downregulate" one word. In Figure 2D, Please use % viability for PBMCs to compare with leukemia cells in Figures 2A-C. Also you can add the word "Differential" at the start of Figure 2 and Figure 3 legends, line 106 and 119. In line 241, please replace "growth inhibitory" with "colonization" and ' in vivo" with "ex-vivo". One of the future outcomes of this study is to validate the ATPase ion channel target of palytoxin as a possible cancer target and future design of simplified analogs based on its essential pharmacophores. 

Author Response

Reviewer 2

Authors did a good job in making a good and important story about the the polyether phycotoxin palytoxin. The study is relevant to marine natural products and aims to highlights the differential and mechanistic cytotoxicity of title molecule against several AML and solid tumors.

We thank the reviewer for the overall positive evaluation of our manuscript and the interesting comments that improved our manuscript.

Authors, please consider changing the title by replacing the word "compound" with "polyether phycotoxin", explicitly describing the chemical class of the title marine natural product.

Done.

Authors, please change "phosphorylation" in line 35 to "dephosphorylation".

Done.

I suggest describing the zebrafish colonization model as ex-vivo rather than in vivo since the treatment conducted outside the animal and not direct systemic treatment after xenografting.

We agreed and changed the text to:

Line 254: “an ex vivo zebrafish xenotransplantation model”

Line 49, please make cis/trans in italics font and add the word "geometrical" before them.

Done.

I suggest to make the word "downregulate" one word.

Done.

In Figure 2D, Please use % viability for PBMCs to compare with leukemia cells in Figures 2A-C.

We thank the reviewer for this remark. We now reported the PBMC results in “Viability (%)” to uniformize the results.

Also you can add the word "Differential" at the start of Figure 2 and Figure 3 legends, line 106 and 119.

Done.

In line 241, please replace "growth inhibitory" with "colonization"

We preferred to replace this term with “anticancer potential”

and ' in vivo" with "ex-vivo".

Done.

One of the future outcomes of this study is to validate the ATPase ion channel target of palytoxin as a possible cancer target and future design of simplified analogs based on its essential pharmacophores.

We thank the reviewer for this suggestion. We added the following discussion:

“One of the future outcomes of this study is to validate the Na+/K+-ATPase inhibition by palytoxin as the possible cancer target. In this context, functional studies (i.e., silencing or overexpressing the pump subunits) and the future design of simplified analogs based on essential pharmacophores of palytoxin might be considered. Palytoxin abrogated tumor formation in a zebrafish xenograft model at concentrations between 10 and 30 pM. We plan to extend our investigation to animal models other than zebrafish to exclude any intrinsic evolutionary resistance developed by aquatic animal models. Besides, studies on primary cells will consolidate the documented differential cytotoxicity”.

Reviewer 3 Report

The study by Kim et al. investigate the mechanisms of cell death induced by a toxic marine compound palytoxin in leukemia cell lines and in zebrafish embryos. The study is solid and well-presented, however, I have the following concerns:

1)    What is the overarching goal of this study? The authors motivate their study in the introduction by stating that “Further characterization of the mode of action of this compound would allow a better understanding of the molecular mechanisms triggered by palytoxin at the cellular level to investigate further the role of different types of signaling mechanisms involved in carcinogenesis”I do not see how this study addressed this – the authors describe pretty much classic apoptosis in leukemia cell lines. Does palytoxin have a potential to become a drug, considering its not drug-like structure? Or is it a tool to study apoptosis? This question was not addressed neither in the intro nor in the discussion. It is important to state this because otherwise it is not clear why this is interesting. 

2)    Since the authors study apoptosis, they should really use a different method than Trypan blue. Annexin V/PI staining is one good alternative because it allows to assess both apoptosis and necrosis.

3)    Is ‘viability’ in Fig 2A-C and ‘cell death’ in Fig 2D the same? Figure legend states ‘viability’ for figure 2D. 

4)    How many donors were PBMCs drawn from? Were they fresh or previously frozen? Here an alternative to Trypan blue should definitely be used as PBMCs and cell lines are very different in size, and it might be hard to distinguish apoptotic cells using Trypan. 

5)    Fig 4D is not marked.

6) Use other human or mouse primary cells to state the outstanding selectivity of the compound. There might be evolutionary reasons for zebrafish being resistant.

Author Response

Reviewer 3

The study by Kim et al. investigate the mechanisms of cell death induced by a toxic marine compound palytoxin in leukemia cell lines and in zebrafish embryos. The study is solid and well-presented, however, I have the following concerns:

We thank the reviewer for the overall positive evaluation of our manuscript and the interesting comments that improved our manuscript.

1)    What is the overarching goal of this study? The authors motivate their study in the introduction by stating that “Further characterization of the mode of action of this compound would allow a better understanding of the molecular mechanisms triggered by palytoxin at the cellular level to investigate further the role of different types of signaling mechanisms involved in carcinogenesis”. I do not see how this study addressed this – the authors describe pretty much classic apoptosis in leukemia cell lines. Does palytoxin have a potential to become a drug, considering its not drug-like structure? Or is it a tool to study apoptosis? This question was not addressed neither in the intro nor in the discussion. It is important to state this because otherwise it is not clear why this is interesting. 

We agree with the reviewer that this formulation in the introduction section is misleading and does not correspond to the results presented.

The aim of this manuscript was to provide a proof-of-principle for the anticancer potential of palytoxin at low picomolar concentrations. At these concentrations, the compound triggers apoptotic cell death mechanisms in both hematological and solid tumor cell models. We used a multi-technique approach both in vitro, by colony formation, and by zebrafish xenograft tumor formation assays to validate the compound’s anticancer potential despite its high molecular weight.

At high nanomolar doses, palytoxins can indeed trigger inflammation, act as a tumor co-promotor, and contribute to carcinogenesis via MAP kinase signaling. We moved this section to the discussion to contrast the compound's effect at high vs. low doses.

Altogether, we edited the introduction and discussion sections to match the text with our intentions and focus on the results of the present manuscript.

2)    Since the authors study apoptosis, they should really use a different method than Trypan blue. Annexin V/PI staining is one good alternative because it allows to assess both apoptosis and necrosis.

We agree with the reviewer. We used trypan blue staining as a screening approach to compare the impact of palytoxin on cancerous and non-cancerous cell models before undertaking mechanistic investigations to analyze the type of cell death induced by palytoxin in selected cell models by different technical approaches. At this first level, it was not our intention to detect the presence of apoptotic cells, an analysis that goes beyond the potentialities of the trypan blue assay.

Subsequently, for the selected cell lines, we quantified apoptotic/necrotic cells by Hoechst/propidium iodide (PI) co-staining and subsequent analysis of nuclear morphology according to a robust method, well accepted in the field (see our manuscripts: Song et al. Mar Drugs (2021); Ha et al.; Mar Drugs (2020); Song et al., Cell Death Dis (2020); Mazumder et al., Cancer Lett. (2018); Cerella et al., Leukemia (2017); Cerella et al., Cell Death Dis (2015)). To this end, images of treated/untreated co-stained cells were captured. Nuclei with a typical apoptotic nuclear condensation/fragmentation phenotype were counted based on images taken from random fields (see Section 4.7 in Materials & Methods).

This approach was performed in parallel with the monitoring of caspase cleavage/activation, confirming the occurrence of apoptotic cell death. In future studies, we will consider AnnexinV/PI assays to characterize different cell death modalities and thank the reviewer for this advice.

3)    Is ‘viability’ in Fig 2A-C and ‘cell death’ in Fig 2D the same? Figure legend states ‘viability’ for figure 2D.

We thank the reviewer for this remark. Now, Figure 2D represents the results as “Cell viability (%)”.

4)    How many donors were PBMCs drawn from? Were they fresh or previously frozen?

Peripheral blood mononuclear cells were purified from three freshly collected buffy coats from healthy adult human donors (Red Cross, Luxembourg) after ethical approval. See the Material and Methods section for details.

Here an alternative to Trypan blue should definitely be used as PBMCs and cell lines are very different in size, and it might be hard to distinguish apoptotic cells using Trypan. 

We thank the reviewer for this comment. As mentioned above, we selected trypan blue assay as the first screening approach to compare the impact of palytoxin in cancerous and non-cancerous cell models.

In our hands, both lymphoid and myeloid cell components of PBMCs are visible and countable using a Malassez counting chamber method and the trypan blue solution, which makes this test robust for the intended purposes. We will surely consider specific and suitable assays for future investigations whenever we plan to characterize and quantify specific types of cell death in PBMCs or primary cells.

5)    Fig 4D is not marked.

We apologize for this oversight. Now, Fig. 4D is marked in the text.

6) Use other human or mouse primary cells to state the outstanding selectivity of the compound. There might be evolutionary reasons for zebrafish being resistant.

We thank the reviewer for this insightful remark. In the next step, we will expand cytotoxicity assays in animal models other than Zebrafish to exclude the intrinsic resistance of aquatic animals. We added here some considerations to the discussion.

Round 2

Reviewer 3 Report

ok